# Is Lipid Specificity Key to the Potential Antiviral Activity of Mouthwash Reagent Chlorhexidine against SARS-CoV-2?

**DOI:** 10.3390/membranes12060616

**Published:** 2022-06-14

**Authors:** Arun K. Rathod, Chetan S. Poojari, Moutusi Manna

**Affiliations:** 1Applied Phycology and Biotechnology Division, CSIR Central Salt & Marine Chemicals Research Institute, Bhavnagar 364002, India; rathod.csmcri21j@acsir.res.in; 2Theoretical Physics and Center for Biophysics, Saarland University, 66123 Saarbrücken, Germany

**Keywords:** chlorhexidine, viral lipid membrane, pore formation, free-energy

## Abstract

Chlorhexidine (CHX), a popular antibacterial drug, is widely used for oral health. Emerging pieces of evidence suggest that commercially available chlorhexidine mouthwash formulations are effective in suppressing the spread of SARS-CoV-2, possibly through destabilization of the viral lipid envelope. CHX is known for its membrane-active properties; however, the molecular mechanism revealing how it damages the viral lipid envelope is yet to be understood. Here we used extensive conventional and umbrella sampling simulations to quantify the effects of CHX on model membranes mimicking the composition of the SARS-CoV-2 outer lipid membrane as well as the host plasma membrane. Our results show that the lipid composition and physical properties of the membrane play an important role in binding and insertion, with CHX binding favorably to the viral membrane over the plasma membrane. Among the simulated lipids, CHX preferentially binds to anionic lipids, PS and PI, which are more concentrated in the viral membrane. The deeper and stable binding of CHX to the viral membrane results in more pronounced swelling of the membrane laterally with a thinning of the bilayer. The overall free energies of pore formation are strongly reduced for the viral membrane compared to the plasma membrane; however, CHX has a larger concentration-dependent effect on free energies of pore formation in the plasma membrane than the viral membrane. The results indicate that CHX is less toxic to the human plasma membrane at low concentrations. Our simulations reveal that CHX facilitates pore formation by the combination of thinning the membrane and accumulation at the water defect. This study provides insights into the mechanism underlying the anti-SARS-CoV-2 potency of CHX, supporting its potential for application as an effective and safe oral rinse agent for preventing viral transmission.

## 1. Introduction

Severe acute respiratory syndrome coronavirus-2 (SARS-CoV-2), a novel coronavirus first identified in December 2019 in Wuhan, China, has spread rapidly and inevitably across the globe resulting in the ongoing coronavirus disease 2019 (COVID-19) pandemic [1,2]. In recent years, mankind has witnessed the emergence of deadly viruses like SARS (in 2002) and the Middle East respiratory syndrome (in 2012), but they were less widespread and contagious than SARS-CoV-2 [3]. SARS-CoV-2 is a single-stranded RNA enveloped virus that is mainly spread through respiratory and oral routes due to the inhalation of virus-laden secretions such as saliva (viral load up to 91.7%), respiratory aerosols, and droplets produced during exhalation, i.e., when an infected person sneezes, talks, or coughs [4,5].

In common with many viruses, SARS-CoV-2 is wrapped in a fatty layer, called the “lipid envelope”, which is studded with proteins that allow the virus to bind and invade the host cell [6]. Inside the host cell, viruses associate with organelle membranes, which is essential for viral replication [7]. The specific composition of the SARS-CoV-2 viral envelope is yet to be determined, but it is considered to be the same as the lipid composition of the endoplasmic reticulum–Golgi intermediate compartment (ERGIC), where the virus buds [8,9]. Interfering with the viral lipid envelope is a widely accepted virucidal strategy to target many coronaviruses, as they are highly sensitive to reagents that disrupt their outer lipid membrane [10,11,12]. In the fight against SARS-CoV-2, one of the main precautionary measures recommended by the World Health Organization (WHO) is frequent cleaning of hands with soap or alcohol-based hand sanitizers [13,14], both agents act by rupturing the viral lipid envelope. Alcohol is also highly efficient at inactivating enveloped viruses, including SARS-CoV-2, on inanimate surfaces/fomites [11]. While these are proven ways of surface neutralization of SARS-CoV-2, the oral antiviral strategies are relatively less explored.

It is broadly accepted that the throat and salivary glands are the major sites of virus replication and shedding in early COVID-19 disease [15,16]. SARS-CoV-2 is detectable from the saliva of infected individuals without or with mild symptoms [15]. Viral load peaks during the first week after symptom onset with the highest potential of viral transmission in the early stages of COVID-19 [17,18]. A few recent studies suggest that oral rinsing should be considered as a potential way to restrict the transmission of SARS-CoV-2 [12,19,20]. Commercially available mouthwashes from various countries have shown efficacy in suppressing bacteria and viruses in the oral cavity and dental aerosols [21,22]. Widely available dental mouthwash components with potent antibacterial, antivirus, and antiseptic activities include ethanol, essential oils, hydrogen peroxide (H_2_O_2_), povidone-iodide (PVP-I), cetylpyridinium chloride (CPC), and chlorhexidine (CHX) [12,23,24,25,26,27]. The proposed effects of mouthwashes in reducing salivary SARS-CoV-2 viral load are based on the assumption that the active components in mouthwashes could destroy the lipid envelope of the virus. The idea is supported by already published data that mouthwashes can inactivate enveloped viruses, including coronaviruses, in the laboratory and in humans, with the likely mechanism being damage to the viral lipid envelope [12,23,24,25,26,27]. 

Emerging pieces of evidence demonstrate the virucidal activity of commercially available oral rinses against SARS-CoV-2 that encourage significant clinical research [19,20,27,28]. PVP-I and CHX appear to be very effective mouthwash reagents in reducing the viral load of SARS-CoV-2 *in vitro* and *in vivo* [28,29,30,31,32,33,34]. An *in vivo* study using a randomized-controlled clinical trial demonstrates that 0.2% CHX and 1% PVP-I oral solutions are effective preprocedural mouthwashes against salivary SARS-CoV-2 in dental treatments [30]. Yoon et al. evaluated the *in vivo* efficacy and showed that 0.12% CHX mouth rinse for 30 s is effective in reducing SARS-CoV-2 viral load in COVID-19 positive patients for 2 h [34]. An *in vitro* study by Jain et al. also reported that 0.2% CHX inactivates more than 99.9% SARS-CoV-2 viruses in a minimal contact time of 30 s and is recommended by the authors to have better efficacy than PVP-I [32]. It is also critical to consider the cytotoxic effects of oral rinses on the host cells prior to assessing their antiviral activities [27]. Xu et al. found that Colgate peroxyl (H_2_O_2_) and PVP-I exhibit very high cytotoxicity, while diluted Listerine (essential oil and ethanol) and CHX (at a concentration mimicking their actual use), exhibited no cytotoxic effect and may thus be considered as good candidates to reduce the spread of SARS-CoV-2 [27].

CHX is a well-established antibacterial agent that exhibits potential virucidal activity against SARS-CoV-2 based on the current knowledge. It is known to be effective against enveloped viruses [24], whose mode of action is thought to be disruption of the cell membrane. CHX is a bisbiguanide compound, a class of compounds known for their bactericidal properties [35,36]. The cationic nature makes it extremely interactive towards negatively charged microbial surfaces. While CHX has been widely exploited for its antimicrobial activity, the mechanisms by which it is taken up by a cell and disrupts/deforms a specific cell membrane are yet to be fully understood. In complementary to experiments, molecular dynamics (MD) simulation emerges as a powerful technique to study the drug-membrane interactions at atomistic resolution [37,38,39]. Recent studies provided mechanistic insights into itraconazole drug-induced pore formation and also quantified the free energies of membrane pore formation [38]. Previous MD simulation studies provide valuable insights into how CHX molecules, at different concentrations and protonation states, are partitioned into and interact with lipid membranes [36,40]. However, these studies focused on simplified model membranes composed of 1,2-dimyristoyl-sn-glycero-3-phosphocholine (DMPC) lipids, while CHX in complex biological membranes may respond very differently. Biological membranes are composed of a large number of lipid types and the lipid composition largely varies among organisms and from one cell type to another [41].

In the present work, we employed extensive atomistic classical MD simulations to study the interactions of dicationic CHX with membranes mimicking the composition of SARS-CoV-2 viral envelope as well as human plasma membrane based on the available lipidomics data [42,43]. Here, the plasma membrane (PM) is modelled on the lipid composition of the human erythrocyte plasma membrane and the SARS-CoV-2 viral membrane (VM) is modelled on the lipid composition of the endoplasmic reticulum (ER) membrane, as suggested by recent studies [42,43]. The present work is focused on deciphering the differential impacts of CHX on these model membranes. We performed biased umbrella sampling simulations to derive the thermodynamics and energetics associated with the CHX-membrane interactions and pore formation, towards understanding the mechanism of CHX-induced membrane damage.

## 2. Materials and Methods

### 2.1. Chlorhexidine Structure and Parameter

As depicted in Figure 1A, CHX is a symmetric molecule with two biguanide (polar, hydrophilic) groups and chlorophenyl (lipophilic) rings connected through a hexane (hydrophobic) bridge. The compound is strongly basic and has multiple protonation states. At pH levels above 3.5, CHX is likely dicationic with positive charges on either side of the hexane linker [35,36]. The initial structure of +2 charged CHX was constructed using the Avogadro tool [44] and the parameters compatible with CHARMM36 force field were obtained from previous studies of CHX in lipid membranes [36]. The full details on CHX parameterization are described elsewhere [36].

### 2.2. Molecular Dynamics Simulations

#### 2.2.1. Simulation of Plasma and Viral Membranes

The initial membranes mimicking the plasma membrane (PM) (Table 1) and viral membrane (VM) (Table 2) were prepared using the CHARMM-GUI server [45,46], and the lipid composition was based on the recent lipidomic studies [42,43]. The lipids [47,48], ions, and TIP3P water model [49] were described using all-atom CHARMM36 force field parameters. The membrane patches obtained from CHARMM-GUI were solvated with the TIP3P water model and neutralized with NaCl ions (system details are provided in Table 3). Next, the membranes were energy minimized to remove any atomic clashes, followed by several short equilibration runs. During equilibration, a temperature of 310 K was regulated using a Berendsen thermostat with a time constant of 1.0 ps and the reference pressure of 1 bar was maintained semi-isotropically using the Berendsen barostat with a time constant of 5.0 ps [50]. The coulombic and van der Waals interactions within 1.2 nm were computed using the Particle Mesh Ewald [51] and cutoff methods respectively, and H-bonds were constrained using the Linear Constraint Solver (LINCS) algorithm [52]. For the final production simulations, the settings were identical, except for switching to the v-rescale (stochastic velocity rescaling) thermostat [53] and the Parrinello–Rahman barostat [54]. The production runs were carried out for 1 microsecond with a time step of 2 fs and output trajectory and energy files were written every 100 ps. The simulations were performed using the GROMACS 2020.2 package [55].

#### 2.2.2. Conventional CHX-Membrane Simulations

The final structure at 1 microsecond from PM and VM simulations served as a starting structure to investigate CHX-PM/VM membrane interactions. We first removed all the water molecules and ions from the final structure, placed CHX molecules (1:40 CHX:Lipid ratio) >2 nm above the membrane surface, and resolvated the system with TIP3P water molecules. The solvated system was neutralized with counter ions and an additional 150 mM NaCl was added to mimic the physiological salt concentration. The rest of the simulation steps and parameters were same as discussed in Section 2.2.1. Each system was equilibrated for 1 microsecond with three repeats. The MD simulation details are given in Table 3.

### 2.3. Potential of Mean Force (PMF) Calculations

#### 2.3.1. Free-Energy of CHX Binding to Membrane

To estimate the free-energy of CHX binding to PM and VM, we carried out umbrella sampling simulations along the reaction coordinate defined by the center of mass distance between CHX and phosphorus (P) atoms of the upper leaflet of the membrane. We first carried out a conventional simulation (1 microsecond) with a single CHX molecule placed above the membrane surface and allowed the drug to spontaneously insert into the membrane. The final structure with CHX inserted into the membrane was used as a starting structure for umbrella sampling simulations. The starting windows/frames for umbrella sampling simulations were generated in two stages. First, from the spontaneously inserted position, the drug was further pulled into the membrane by 0.5 nm (in 50 ns) using a force constant of 4000 KJ mol^−1^ nm^−2^ and later pulled away from the membrane surface by 3.3 nm (in 250 ns) with a force constant of 1000 KJ mol^−1^ nm^−2^. The pull rate was set to 1.10–5 nm nm^−1^ in both stages. The center of mass pulling was carried out using umbrella potential with a cylinder geometry with a radius of 2.5 nm applied to P atoms of the upper leaflet. The rest of the parameter settings were the same as those used during conventional production simulations. The umbrella sampling simulation in each of the 33 windows (spaced by 0.1 nm) was carried out for 200 ns and the last 150 ns was used for analysis. The PMFs were calculated using gmx wham code and the errors were estimated using the bootstrap method [56].

#### 2.3.2. Free-Energy of Pore Formation

The free-energy of pore formation was computed along the reaction coordinate termed “chain coordinate” ξ_ch_, which defines the connectivity of polar atoms using a dynamic membrane-spanning virtual cylinder [57,58]. Depending on the membrane thickness, the cylinder is split into N_s_ slices and each slice is occupied by polar heavy atoms, thus ensuring a continuous transmembrane defect. The ξ_ch_ is unitless and ξ_ch_ = 0.25 indicates unperturbed flat membranes, whereas ξ_ch_ = 1 indicates a continuous polar defect. The ξ_ch_ has been extensively used to study pore formation in lipid membranes and more recently to study the effect of drugs on pore formation [38]. Following the protocol as used in an earlier study [38], the cylinder was defined with a radius of 1.2 nm and decomposed into 28 and 37 slices for PM and VM, respectively, with a thickness of 0.1 nm for each slice. For polar atom connectivity, we choose oxygen atoms of water and oxygen atoms of phospholipid molecules (oxygen atoms of phosphate and carbonyl oxygen atom of ester). The PMFs were computed using the umbrella sampling method. The initial frames for umbrella sampling simulations were extracted by pulling the system along the “chain coordinate” ξ_ch_, within 100 ns. For both the membrane systems, 27 windows were used, with a force constant of 5000 kJ mol^−1^ for windows at ξ_ch_ < 0.7 and 10,000 kJ mol^−1^ at ξ_ch_ 0.7–1.0. The rest of the simulation parameters were identical to the conventional simulations described above. Each window was simulated for 250 ns and the PMFs were obtained using the last 200 ns with the gmx wham module and errors estimated using the bootstrap method [56]. These simulations were performed with varying CHX:Lipid ratios (0:40, 1:40 and 3:40) to understand the concentration-dependent effects on pore formation.

## 3. Results

### 3.1. CHX is More Stable in the Viral Model Membrane Than in the Plasma Membrane

Our simulations show that CHX molecules spontaneously bind to both the model membranes, but the residence time of CHX varies significantly in these two membranes. CHX molecules initially placed >2 nm above the average phosphate plane, started interacting with the viral membrane (VM) almost immediately and most of these drug molecules immersed into the VM within 10 s of ns (Appendix A). While for the plasma membrane (PM), few CHX molecules insert in the membrane within 50 ns, but the majority continue to oscillate between water and membrane phases (Appendix A). We started our production simulations from the point where all CHX are bound to membranes and the center of mass distribution of individual CHX molecules during the subsequent 1 µs simulations are shown in Figure 1B,C. It is apparent from Figure 1B that once the CHX molecules are partitioned from water to the membrane phase of VM, they remain bound to the membrane for the rest of the simulation time. CHX molecules remained localized well below the phosphate plane of the VM throughout the simulations. While a stable binding was noted in VM, several unbinding and binding events were observed for CHX molecules in PM (Figure 1C). These results clearly suggest that the membrane-binding of CHX is strongly modulated by lipid composition and the physical properties of the membrane.

The selected snapshots of the systems (Figure 2A,B) and the atom-density profiles (Figure 2C,D) demonstrate that the CHX molecules penetrated slightly deeper below the phosphate plane in the VM membranes than in PM. As can be seen in the snapshots, the positively charged biguanide groups of a membrane-bound CHX interact with the lipid headgroups, while the chlorophenol rings are partitioned favorably into the lipid hydrocarbon chain region. A similar orientation of CHX was reported by a previous computational study [36], which validates our results. A more quantitative picture of conformation of CHX and depth-of-penetration of its different functional groups in membranes is presented in Figure 1D,E. Figure 1D convincingly demonstrates a nearly ~2–4 Å deeper penetration of CHX in VM than PM. In VM, the chlorophenyl rings partitioned into the hydrocarbon region with Cl atoms and biguanide moieties reaching an average of ~8.5 Å and ~3 Å, respectively, below the P plane. However, in PM, Cl atoms penetrated ~4–5.5 Å below the P plane, which is considerably less than VM. In addition, the biguanide groups in PM are mostly located at the P region. In membranes, CHX adopts a wide range of conformations from compressed to extended structures, defined here by the distance between the chlorine atoms (d_Cl-Cl_) at the two ends of the molecules (Figure 1E). The Cl-Cl distance distribution shows a peak at d_Cl-Cl_ = 1.4 nm in VM, corresponding to a CHX structure like a “two-pronged fastener” with Cl atoms of the chlorophenyl rings pointing down to the bilayer core and guanide groups as lipid headgroup anchors. In VM, the d_Cl-Cl_ distribution is a little broader (~0.9 to 1.3 nm) with a greater population of slightly more compressed structures, corresponding to “wedge” shape structures of CHX. CHX with fully extended structures are less abundant in both membranes.

In the present work, we found no major affinity of CHX molecules to self-aggregate in the presence of lipid membranes (Appendix A, top-views). CHX–CHX interactions are slightly more favored in the presence of PM than VM membranes (Appendix A). However, these interactions are mainly transient in nature and no persistent CHX aggregates were detected due to repulsion of positively charged CHX molecules, while interacting favorably with negatively charged lipids.

### 3.2. CHX has a Stronger Binding Affinity to the Viral Membrane

To get a quantitative estimation of the interactions of CHX with the model membranes, we calculated interaction energies of CHX with different lipid types in each membrane (Figure 3A,B). It is evident from our results that CHX has much stronger binding interactions overall with VM than with PM. For the VM (Figure 3A), the zwitterionic DOPC lipids, which are the most abundant VM lipids (35% of the total lipid), give the highest contribution to the total CHX-lipid interaction energies, followed by DOPE (20% of the total lipid) and POPC (15% of the total lipid) lipids. Interestingly, SAPI lipid, although a minor component (10% of the total lipid) of VM, significantly interacts with CHX (Figure 3A). Indeed, from the normalized interaction energy plot (inset plot in Figure 3A), it can be seen that the negatively charged SAPI lipids exhibit the strongest affinity to CHX, followed by another anionic lipid POPS. The binding preference is possibly driven by the electrostatic attraction between the negatively charged lipid headgroups and the positively charged CHX molecules. The binding of CHX with lipids are further stabilized by the formation of hydrogen bonds between biguanide groups of CHX and polar lipid head groups (Figure 3C,D). We found that SAPI lipids also have the highest affinity for being hydrogen-bonded with CHX, followed by POPS in VM (Figure 3C).

The plasma membrane has a distinctly different lipid composition as compared to VM. PLPC and PSM, being the two most abundant phospholipids of the model PM membrane (15% and 12% of total lipid, respectively), contribute the most to the total interaction energies with CHX (Figure 3B). Unlike the viral membrane, cholesterol is a major component of PM and interacts with CHX mainly through vdW interactions with its steroid ring. However, the preference to interact with CHX is strongest for anionic lipids, SAPI and SAPS (inset plot in Figure 3B). These two anionic lipids also show increased hydrogen bonding preference for CHX, as compared to zwitterionic lipids (Figure 3D). In general, the ER-Golgi membrane from where the viral membrane buds, is more anionic in nature than the plasma membrane. In the model systems studied here, the total concentration of anionic lipids in the viral membrane is 15%, while they are only a minor component (total concentration is 4%) in the plasma membrane. Our results clearly demonstrate that CHX has greater preference to bind lipids with anionic headgroups such as PI and PS (Appendix A) and that higher abundance of these lipids in the viral membrane causes stronger and longer binding of CHX compared to the plasma membrane.

### 3.3. Free-Energy of CHX Binding to PM and VM

In Figure 4A we show the free energy of CHX binding to PM and VM as a function of distance from the water phase as computed by umbrella sampling simulations. The free-energy profiles show existence of a barrier at the membrane-water interface (z = 1.7 nm) for CHX in PM, which indicates that spontaneous membrane insertion is resisted by the PM, whereas in VM, CHX inserts spontaneously without any barrier. Other than the initial barrier difference at the membrane-water interface, there is also a difference in the positions of the local minima. The free-energy minima is shifted slightly deeper into the hydrophobic core of the membrane for CHX in VM than in PM, which indicates that CHX prefers to be localized at the membrane-water interface in PM, but immersed deep into VM. These simulations show the effect of the lipid composition and the membrane property on CHX binding, and the free-energy of binding to VM is −27 kJ/mol, while that to PM is −18 kJ/mol (Figure 4A). The results suggest that, as compared to PM, the binding to CHX is more favorable in VM.

### 3.4. Membrane Perturbing Effects of CHX on Plasma vs. Viral Membrane

Next, we examined the effect of CHX on the physical properties of the model membranes. To quantify the impact of CHX, the results were compared with the respective membrane-only systems. As expected, the plasma membrane model is significantly thicker and more ordered than the viral membrane (Figure 5, Appendix A), due to a higher level of cholesterol and lower degree of lipid acyl chain unsaturation. We observed that the plasma membrane is less affected by CHX binding than the viral membrane. The atom density profiles demonstrate that the binding of CHX causes a shift in the peak position of the phosphate atoms more towards the bilayer center in the case of VM, but no apparent change is noted for PM (Figure 2C,D). The results suggest that the long-term binding of CHX with VM causes an overall thinning of the bilayer. The top view of the system can be obtained from the thickness map (Figure 5). It can be viewed clearly that many thinner regions appear in the viral membrane after CHX binding (Figure 5A–C). Interestingly, the thinner regions in the VM overlap with the regions with high CHX local density, but no such pattern nor any appreciable thinning by CHX is noted for PM (Figure 5D–F). In comparison, the VM membrane without CHX is nearly homogeneous in nature, while PM due to its high cholesterol content shows lateral heterogeneity in terms of the lateral organization of lipids (Figure 5F). Our results are in agreement with previous studies reporting the nanoscale heterogeneity in the plasma membrane, which is known to be associated with the biological functions [41,59,60]. Further, we also observed swelling of the membrane on CHX binding, which is evident from the increase in the area-per-lipid (APL) from ~61.51 Å^2^ to ~62.81 Å^2^ in VM and from ~43.53 Å^2^ to ~44.17 Å^2^ in PM (Appendix A). Due to the deeper penetration of CHX molecules in the viral membrane, the effect is again more pronounced in the viral membrane (ΔAPL = 1.30) than in the PM membrane (ΔAPL = 0.64). CHX also causes local disordering of the lipid acyl chains in both of the model membranes. As can be seen from Appendix A, lipids that are strongly bound with CHX molecules are less ordered as compared to the respective model membrane without CHX. Here the disordering effect on CHX bound-lipids are more pronounced in PM than VM. The more fluid-like nature of VM compared to PM may facilitate the deeper penetration and better accommodation of the drug molecules in the former membrane. Overall, our results suggest that CHX modulates the physical properties of both membranes. Because of the stable binding, CHX has stronger effects on the viral membrane, in terms of membrane thinning or an increase in membrane area.

To quantify the effect of CHX on pore formation, we computed the free energies of pore formation over PM and VM containing CHX in the CHX:Lipid ratios of 0:40, 1:40 and 3:40. The simulation snapshots taken at ξ_ch_ = 1 for VM and PM are shown in Figure 6. The PMFs reveal stark differences between PM and VM to facilitate membrane thinning and pore formation (Figure 4B). In the absence of CHX, the free-energy required to facilitate a thinned membrane (at ξ_ch_ = 0.75) is 135 kJ mol^−1^ for PM and 69 kJ mol^−1^ for VM, and the free-energy of pore formation (ΔG_pore_, ξ_ch_ = 1) is 217 kJ mol^−1^ for PM and it is reduced by ~50% to 110 kJ mol^−1^ for VM. The PMFs for pure membranes indicated that the lipid composition of VM favors pore formation over PM. However, addition of CHX to PM leads to a large reduction in free energy of the open pore ΔG_pore_ in a concentration-dependent manner. Namely, at a CHX:Lipid ratio of 1:40, CHX reduces ΔG_pore_ by ∼25 kJ mol^−1^, while at 3:40, CHX leads to a large reduction in ΔG_pore_ by ∼66 kJ mol^−1^. In contrast, the effect of CHX on VM is moderate and reduces the ΔG_pore_ by only ∼10 kJ mol^−1^ at a CHX:Lipid ratio of 3:40.

The two likely mechanisms for the CHX effect on the free energies of pore formation are, first, by altering the physical properties of the membrane, and second, by interactions of CHX with the water defect. As evident from the area-per-lipid and membrane thickness analysis, insertion of CHX into the membrane increases the membrane area and renders the membrane thinner, thus facilitating pore formation. The findings are consistent with previous studies that showed that membrane thickness directly correlates with the free energies of pore formation [57,58,61,62]. Further, visual inspection (Figure 6) revealed binding of CHX to the water defect. In conventional simulations, CHX orients parallel to the surface, with the positively charged biguanide groups interacting with the lipid headgroups, while the chlorophenol rings make contact with the hydrophobic lipid tails. During membrane thinning, CHX diffuses towards the emerging water defect, thus rationalizing the reduced free energy observed at ξ_ch_ = 0.75. At the water defect, CHX reorients such that the charged biguanide groups interact with water and the hydrophobic chlorophenol rings point towards the lipid tails. However, CHX is not strongly enriched at the defect due to favorable interactions with the headgroups of PS and PI lipids (Appendix A). These findings indicate that free energies of pore formation depend on a combination of altering membrane properties and diffusion of CHX towards the water defect. The large reduction in free energy observed for PM is likely due to membrane perturbation by the inserted CHX, which is relieved on thinning of the membrane and pore formation.

## 4. Discussion

The present study provides atomistic insights into the interactions of the antibacterial drug CHX with model membranes mimicking the SARS-CoV-2 viral envelope lipid membrane (VM) and the host plasma membrane (PM), using conventional and umbrella sampling simulations. Our simulations show that CHX molecules quickly attach and immerse into the viral membrane. Whereas its attachment to the plasma membrane is often followed by frequent unbinding/rebinding events and is associated with a free-energy barrier, unlike VM. The free-energy profiles demonstrate that the binding of CHX to viral membrane is more favorable than binding to plasma membrane. The present work suggests that the binding preference of CHX is attributed to the lipid composition and the membrane’s physical properties. The two membranes vary significantly in their lipid compositions. Cholesterol and sphingolipids are abundant components of the PM but are scarce in the VM membrane model, which on the other hand is enriched with anionic and unsaturated lipids [41,42,43,63]. Our simulations show that irrespective of the membrane model, CHX preferentially binds to anionic lipids, PS and PI lipids, through electrostatic and hydrogen-bonding interactions. The viral membrane, being much richer in anionic lipids than the plasma membrane, establishes stronger and stable interactions with CHX. In addition, the two membranes have starkly different physical properties. The VM is fluid in nature, whereas the PM model is much more ordered and rigid. This provides a plausible explanation for our finding that CHX molecules are completely immersed in the hydrophobic region of VM, but can accumulate only at the lipid–water interface of PM.

Earlier study showed that the model membrane composed of DMPC can absorb remarkably high quantity of CHX (30:100 CHX:DMPC) without disruption or any significant change in bilayer structure [64]. But the finding is contradictory to the previously reported lytic and membrane destabilizing properties of CHX [65,66]. One possible reason could be that the simplistic model membranes, such as DMPC, cannot reproduce the membrane perturbing activities of CHX. Therefore, one should consider more realistic model membranes to account for the complexity of the biological membranes. In the present study, we found that CHX exerts only a moderate effect on the structural properties of the plasma membrane, but causes considerable lateral expansion and thinning of the viral membrane. Such drug-induced biophysical changes in the viral membrane possibly affect the ability of the virus to infect the host cell, even in the absence of complete lysis [12,67]. The spike glycoprotein of SARS-CoV-2, which is responsible for initiating the attachment of the virus to its host, is mechanically anchored to the viral membrane through its transmembrane domain. Thus, deformation of the viral membrane may impact the conformation and function of the infectious viral proteins, leading to the inactivation of the virus prior to complete disruption of the viral membrane.

Membrane disruption or lysis is another proposed mechanism by which active oral rinse components neutralize enveloped viruses, including SARS-CoV-2 [12,19,21]. A majority of current knowledge is based on ethanol-based mouthwash formulations. Diluted ethanol reportedly causes swelling and significant interdigitation of model membranes [12,67,68,69]. Interdigitation, which is not observed in our study with CHX, deforms the lamellar bilayer structure and leads to membrane fusion. *In vitro* studies reported leakage of intracellular content when the cell was treated with concentrated ethanol [12,70,71]. Ethanol, being a short-chain alcohol, can permeate model phospholipid membranes through passive diffusion and may induce several non-bilayer structures [68]. Unlike ethanol, our results show that CHX is less likely to spontaneously diffuse to the bilayer center, due to strong interactions with lipid headgroups through the biguanide moiety. Our simulations suggest that CHX facilitates pore formation in the membrane by combination of bilayer thinning and accumulation at the water defect. The PMFs reveal VM is more prone to pore formation than PM, and the free energy of open pore in PM would reduce only at concentrations exceeding the simulated 3:40 (CHX:Lipid) ratio. Our simulations rationalize the experimentally reported effect of CHX on membranes. The present work suggests that CHX not only has a greater affinity to bind the SARS-CoV-2 outer lipid membrane than the host plasma membrane, but it can also cause significant modulation of the biophysical properties of VM even at a low concentration, while having a moderate impact on PM. The finding is in line with previous reports that CHX has low levels of toxicity to mammalian cells, despite its strong antimicrobial activity [27,66,72]. Our results support the recent experimental finding that diluted CHX solutions (0.12% or 0.20%) appear to be very effective in reducing the viral load of SARS-CoV-2, and at the same time safe for human use [27,28,29,30,31,32,33,34]. 

In summary, the present work provides atomistic insights into the differential impacts of CHX on host plasma membranes vs. viral outer lipid membranes, in particular on how lipid composition modulates CHX binding to membranes and perturbs the effects induced by CHX on membrane structure. The results have implications for understanding the mechanism of the potential veridical activity of CHX against SARS-CoV-2, for the application of CHX mouthwash formulation to suppress the spread of COVID-19.

## Figures and Tables

**Figure 1 membranes-12-00616-f001:**
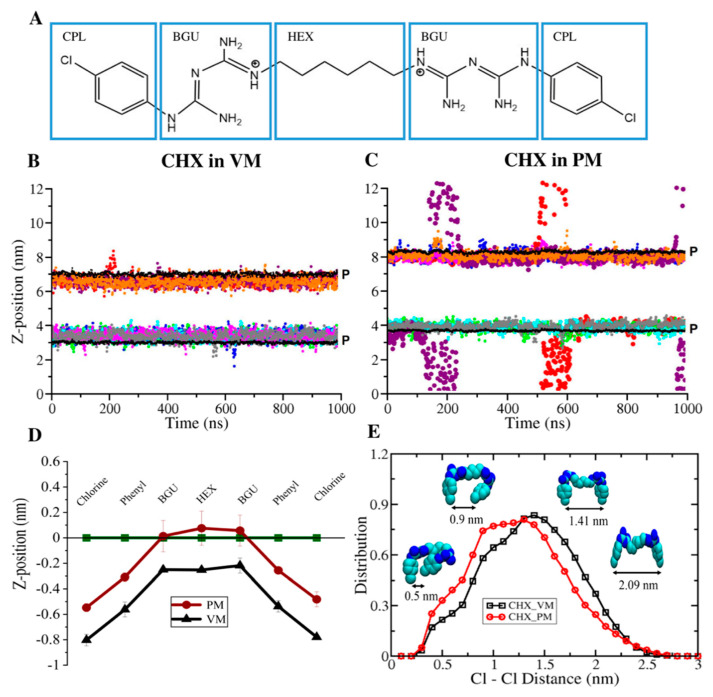
(**A**) Structure of dicationic chlorhexidine (CHX) molecule showing its different functional groups, chlorophenyl (CPL), biguanide (BGU), and hexane (HEX). (**B**,**C**) The center of mass distribution of individual CHX molecules along the normal bilayer, starting from membrane bound CHX. The black lines represent the average planes of P atoms of two bilayer leaflets. Individual CHX molecules are colored separately. (**D**) Z-positions of the functional groups of membrane-bound CHX with reference to the P plane (indicated by line at Z = 0). (**E**) Distribution of ClCl distance of CHX in membranes.

**Figure 2 membranes-12-00616-f002:**
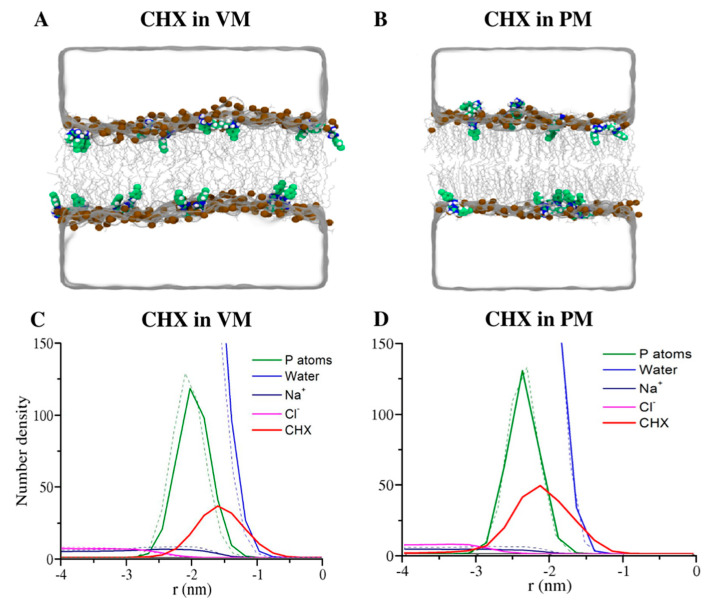
(**A**,**B**) MD simulation snapshots of CHX binding to VM/PM. Upper and lower leaflets of the membrane are represented by lipid phosphorus atoms, which are rendered as brown spheres, lipid tails are shown as gray lines, and water is shown as a transparent surface. Ions are not shown for clarity. (**C**,**D**) Atomic density profiles. The dotted lines represent membrane-only systems, and solid lines represent CHX-containing membranes.

**Figure 3 membranes-12-00616-f003:**
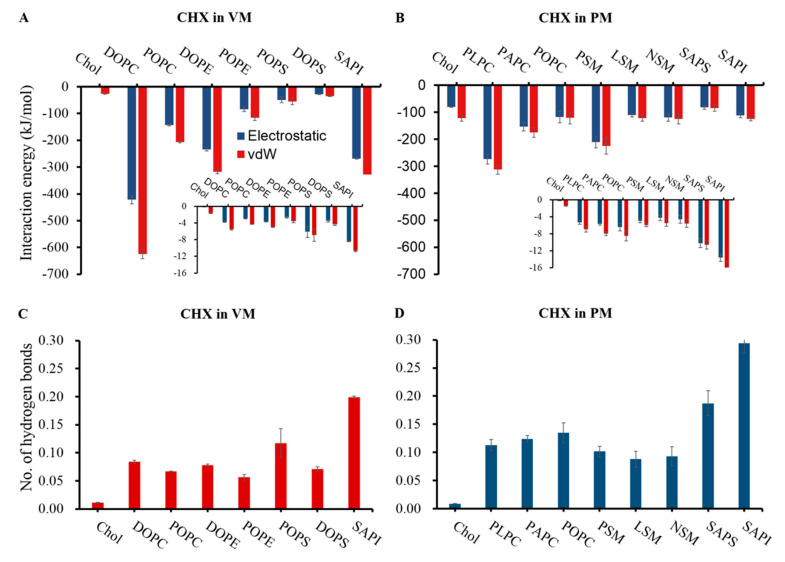
(**A**,**B**) Interaction energies of CHX with each lipid type of the viral membrane and plasma membrane models. Inset plots demonstrate the interaction energies normalized by lipid numbers. (**C**,**D**) Number of hydrogen bonds of CHX with each lipid type of the viral membrane and plasma membrane models, normalized with respect to lipid numbers.

**Figure 4 membranes-12-00616-f004:**
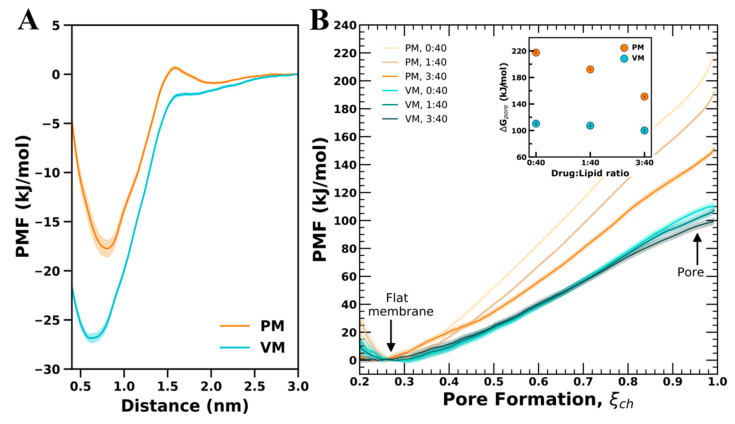
Free-energy of binding and free-energy of pore formation over PM and VM for CHX. (**A**) PMFs showing CHX binding free-energy for VM (cyan shaded) and PM (orange shaded). (**B**) PMFs showing effect of CHX on pore formation, which is dependent on the CHX:Lipid ratio (see legend). The ΔGpore with respect to CHX:Lipid ratio taken at ξ = 1 is shown in the inset.

**Figure 5 membranes-12-00616-f005:**
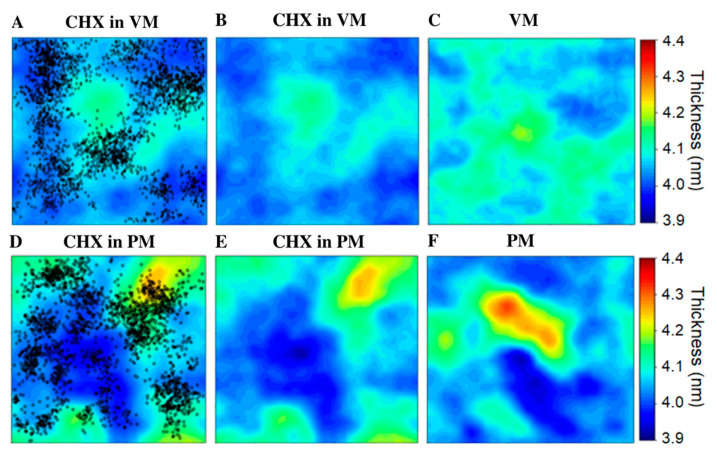
(**A**,**D**) Bilayer thickness of CHX-membrane systems projected on the membrane plane; upper and lower panels correspond to VM and PM, respectively. The X-Y positions of CHX molecules are shown as black dots. In (**B**,**E**), CHX are removed for clarity. (**C**,**F**) Thickness maps of membrane only systems, without CHX.3.5. Effect of CHX on Pore Formation.

**Figure 6 membranes-12-00616-f006:**
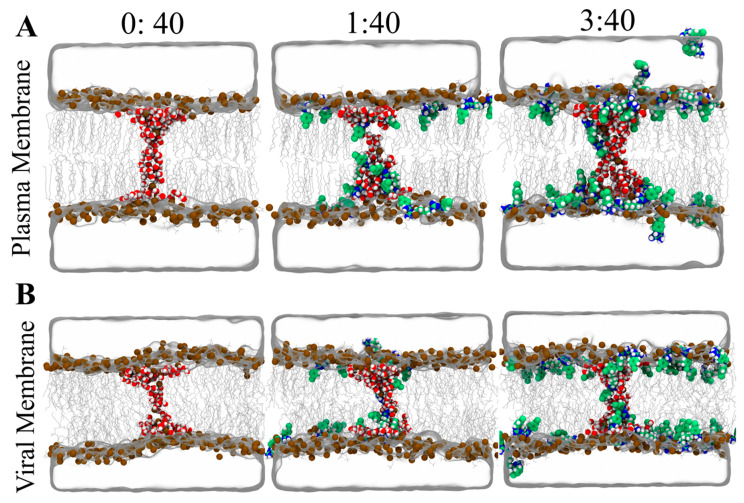
MD simulation snapshots of pore formation at ξ = 1 for CHX:Lipid ratios 0:40, 1:40 and 3:40. (**A**) Pore formation snapshots in PM and (**B**) Pore formation snapshots in VM. Figures are rendered as described in Figure 2A,B.

**Table 1 membranes-12-00616-t001:** The plasma membrane (outer leaflet) composition used in simulations.

Outer Leaflet Plasma Membrane Composition Used in Simulations		
Lipid	Structure	Mol %
PLPC	PC 16:0–18:2	15
PAPC	PC 16:0–20:4	8
POPC	PC 16:0–18:1	5
PSM	PC 18:1–16:0	12
LSM	PC 18:1–24:0	8
NSM	PC 18:1–24:1	8
SAPS	PS 18:0–20:4	2
SAPI	PI 18:0–20:4	2
CHOL	Cholesterol	40

**Table 2 membranes-12-00616-t002:** The viral membrane composition used in simulations.

Viral Membrane Composition Used in Simulations		
Lipid	Structure	Mol %
POPC	PC 16:0–18:1	15
DOPC	PC 18:1–18:1	35
POPE	PE16:0–18:1	10
DOPE	PE 18:1–18:1	20
POPS	PS 16:0–18:1	2
DOPS	PS 18:1–18:1	3
SAPI	PI 18:0–20:4	10
CHOL	Cholesterol	5

**Table 3 membranes-12-00616-t003:** All the simulations carried out to study CHX-PM/VM interactions.

Simulation Type	Membrane	No. of CHX	No. of Lipids	No. of Waters	System Size (nm × nm × nm)	No. of Repeats	Total Time (µs)
Conventional	PM	-	320	12,482	8.31 × 8.31 × 10.07	1	1
	VM	-	320	16,401	9.88 × 9.88 × 9.09	1	1
	PM	8	320	18,215	8.35 × 8.35 × 12.50	3	3
	VM	8	320	22,991	10.13 × 10.13 × 10.63	3	3
	PM	1	160	7278	6.27 × 5.43 × 11.13	1	1
	VM	1	160	9598	7.51 × 6.50 × 9.95	1	1
Umbrella Sampling-Binding energy	PM	1	160	7278	6.27 × 5.43 × 11.13	1	6.6
	PM	1	160	9598	7.51 × 6.50 × 9.95	1	6.6
Umbrella Sampling-Pore formation	PM	-	320	11,861	8.40 × 8.40 × 9.59	1	6.75
	PM	8	320	11,877	8.39 × 8.39 × 9.66	1	6.75
	PM	24	320	11,845	8.54 × 8.54 × 9.50	1	6.75
	VM	-	320	13,342	9.97 × 9.97 × 7.98	1	6.75
	VM	8	320	13,342	10.03 × 10.03 × 7.94	1	6.75
	VM	24	320	13,342	10.17 × 10.17 × 7.86	1	6.75

## Data Availability

Simulation files are available on request from corresponding authors.

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
