# Peer review of "Is Lipid Specificity Key to the Potential Antiviral Activity of Mouthwash Reagent Chlorhexidine against SARS-CoV-2?"

_membranes, 2022, doi:10.3390/membranes12060616_

Round 1

Reviewer 1 Report

I would suggest a table where the authors fully describe the Simulated System (no. of atoms, ystem, simulation time, no. of replicates, ...). I raise some concerns about a study where no experimental data is presented. The conclusions are not convincing I would suggest additional, where the authors could use to sets of simulations with diferent temperatures and also use dipalmitoylphosphatidylcholine (DPPC), which serves as a useful model for understanding the physical properties of biological membranes.
What is the partition coefficient of CHX? What about the number of water molecules?
Solubility of CHX could also be calculated.
Simulations could be done for a number of systems indifferent CHX concentrations.

Reviewer 2 Report

This manuscript reports an interesting MD simulations study to understand mode of action of chlorhexidine on viral membrane (SARS-CoV-2) and plasma membrane. MDs demonstrated preferential binding of chlorhexidine on viral membrane. MD studies reported here are quite convincing and suitable for publication in the journal. However, authors should edit their manuscript for better clarity and language. Some rational should be provided about the choice of lipids used. Legend of Fig 6 is not clear. What is the dimension of pore? 
